Accelerated straw decomposition and mitigated methane emissions via autumn puddling incorporation enhances soil health and yield stability in cold-region rice systems of China

He Bing hb@jlnku.edu.cn hebing02358531@126.com 1
Si Yonglin 2
Li Chao 1
Wang Limin 3
1 Agricultural College, Jilin Agricultural Science and Technology University , Jilin , China
2 Jilin Agricultural University , Jilin , China
3 Jilin City Meteorological Bureau , Jilin , China
Shahzad Tanvir
Electronic publication date: 2025 Oct 27
Publication date: 2025
Volume: 13
Electronic Location ID: e20264
Received 2025 Jul 8; Accepted 2025 Sep 29
Copyright: ©2025 He et al.
Copyright year: 2025
Copyright holder: He et al.
License: This is an open access article distributed under the terms of the Creative Commons Attribution License, which permits unrestricted use, distribution, reproduction and adaptation in any medium and for any purpose provided that it is properly attributed. For attribution, the original author(s), title, publication source (PeerJ) and either DOI or URL of the article must be cited.
License URL: https://creativecommons.org/licenses/by/4.0/

Keywords: Rice straw incorporation, Methane emissions, Soil redox potential, Straw decomposition, Cold-region rice cultivation, Yield stability

Funding: The Jilin Provincial Key Research and Development Project 20240203005NC This research was funded by the Jilin Provincial Key Research and Development Project, grant number 20240203005NC. The funders had no role in study design, data collection and analysis, decision to publish, or preparation of the manuscript.

==============================
This study discusses two challenges to rice production in China’s cold growing regions: the slow decomposition of straw due to low temperatures in the winter, and the environmental threats—concentrated methane emissions and soil reduction stress—associated with incorporation in the spring. To address these issues, a field experiment was conducted over two years (2022–2023) in paddy fields in central Jilin Province. The outcomes of three treatments were compared: no straw return (CK), puddling with water to promote straw incorporation in the spring (T1), and puddling with water to promote straw incorporation in the autumn (T2). The results demonstrated that T2 significantly enhanced the straw decomposition rate compared to T1 by extending the decomposition period, resulting in a 2.9% increase in straw weight loss after incorporation and a reduction in total methane emissions by 32.3%. T2 produced a smaller decline in soil redox potential, thereby alleviating soil reduction stress. Compared to CK, T2 significantly increased the levels of nitrogen, phosphorus , and calcium in the soil. T2 also maintained greater root activity and higher stem numbers than T1. While the grain yield under T2 did not differ significantly from that of CK, the yield under T1 was significantly reduced by 12.7%. Thus, incorporating puddling in autumn accelerates straw decomposition, reduces summer methane peaks, alleviates soil stress, enhances nutrient supply, and stabilizes yields. This makes it a sustainable strategy for managing straw in cold-region rice systems.

Introduction

Jilin Province is a major rice-producing region in China. Both the yield and cultivation area in Jilin have demonstrated sustained growth in recent years. By 2023, the rice planting area had reached 828,800.0 ha, with a total yield of 6,820,600.0 t and an average yield of 8,229.3 kg ha−1 (Jilin, 2025). However, the expansion of rice cultivation has intensified a significant challenge: the disposal of large quantities of rice straw (Alengebawy et al., 2023). The primary conventional straw disposal method was open burning, but national environmental policy reforms have made this practice unviable (Li, Miao & Zhang, 2022; Singh et al., 2021).

Alternatives to open burning for rice straw disposal have been widely studied (Singh & Brar, 2021). The most effective and convenient of these methods involves the direct incorporation of straw into the field (Li et al., 2018). This method not only enhances soil nutrient levels and improves the soil environment (Ninkuu et al., 2025); it also increases the nitrogen content across various soil layers (Cui, Zhu & Cao, 2022) and supplements potassium content, leading to a 5.0% increase in rice yield (Li et al., 2023a). Additionally, it reduces emissions of methane, carbon dioxide, fine particulate matter, and other pollutants associated with straw burning (He, Liu & Zhou, 2020; Bhattacharyya et al., 2021).

However, directly incorporating straw also generates substantial methane emissions (Wu et al., 2024). In addition, the decomposition process following incorporation lowers the soil’s redox potential, resulting in highly reducing soil conditions (Liu et al., 2021). This affects rice root activity (Peng et al., 2024) and low-position tillering, ultimately reducing rice yields (Tian et al., 2022). Incorporation in spring is likely to cause concentrated methane emissions and more intense reduction stress (Zhang et al., 2023). Conversely, autumn incorporation in China’s cold-region often leads to an incomplete decomposition of incorporated straw due to the prolonged low winter temperatures, compromising the decomposition’s effectiveness (He et al., 2024).

Previous research has shown that continuous flooding can increase methane emissions by up to 90.0% (Sanchis et al., 2012). Managing water during the fallow winter season can further increase methane emissions by reducing the inhibition of acetotrophic methanogens (Zhang et al., 2013). Studies conducted in California have indicated that due to straw incorporation, methane emissions from paddies maintained under flooded conditions in winter accounted for approximately 50.0% of the total annual emissions (Fitzgerald, Scow & Hill, 2000). Additionally, field trials conducted in northeastern Japan have demonstrated that accelerating the decomposition of rice straw during the off-rice season effectively reduced methane emissions from paddy fields in cold temperate regions during a single rice-growing season (Nakajima et al., 2016). Research in Jiangsu Province, China, confirmed this finding, showing that methane emissions during the fallow season following straw incorporation ranged from 9.6% to 33.1% of the annual total (Zhang et al., 2011). Moreover, research in the rice-growing areas of southern China showed that autumn incorporation increased straw decomposition and nutrient release rates by 8.9% and 109.0%, respectively, compared to spring incorporation (Wang et al., 2022).

However, few studies have examined the impact of this practice on the yields and agronomic traits of rice, and most of these studies have focused on warmer southern regions (Martinez et al., 2018; Martinez et al., 2022; Vo et al., 2018). In reference to the cold rice-growing areas of northern China, an important question thus arises: can crushed straw be incorporated through puddling immediately after the autumn rice harvest by flooding and plowing the fields? This approach could extend the straw decomposition period, potentially reducing concentrated methane emissions during the hot summer period while mitigating high-intensity soil reduction stress. This could help ensure stable rice yields.

To test this hypothesis in this study, we evaluated paddy fields in central Jilin Province, using a locally dominant rice variety as the test cultivar. We comparatively analyzed yields, methane emissions, soil redox conditions, and the rice’s agronomic traits under the influence of different straw incorporation practices. The objectives were to quantify and compare decomposition rates under autumn and spring incorporation, compare seasonal methane emission patterns, evaluate soil redox dynamics and nutrient changes, and assess the impacts on rice agronomy and yield in central Jilin Province. The results can be used to help select the best straw management strategy for ensuring sustainability in cold-region.

Materials & Methods

Experimental site and cultivar

The experiment was conducted at the off-campus experimental station of Jilin Agricultural Science and Technology University in Yongji County, Jilin City, Jilin Province (N43°836′, E126°113′; altitude: 243.0 m). This station experiences a northern temperate continental monsoon climate, with an annual accumulated temperature of 2,790.0 °C, an average annual precipitation of 690.0 mm, a frost-free period of approximately 130 days, and a total sunshine duration ranging from 2,500 to 2,900 h. Three adjacent paddy fields with independent irrigation and drainage systems were selected, each of which had an area of approximately 1,000.0 m2. Within each paddy field, three 20.0 m × 16.7 m plots were established and randomly assigned to treatments using a randomized complete block design. This design accounted for potential soil microvariability across experimental stations. In accordance with the World Reference Base for Soil Resources (Anjos et al., 2015), the soil type in the experimental paddies was classified as Clay Gleysols. The soil’s pH was 5.57, available phosphorus was 23.9 mg kg−1, available potassium was 219.4 mg kg−1, organic matter was 27.8 g kg−1, and alkali-hydrolyzable nitrogen was 168.0 mg kg−1. The experiment was conducted over two years, from April 2022 to December 2023. The rice cultivar used was Jihong 9, a dominant variety in Jilin Province.

Experimental design

Three treatments were established. In the control (CK) treatment, there was no straw return. The rice straw was completely removed from the field after the harvest. Under spring incorporation (T1), the straw was incorporated with water via puddling in the spring, that is, immediately before rice transplanting in the cultivation year (on May 25, 2022, and May 27, 2023). Finally, under autumn incorporation (T2), the straw was incorporated with water via puddling in the autumn, that is, immediately after rice harvesting in the year preceding cultivation (on September 28, 2021, and September 29, 2022).

Based on the average post-harvest straw yield of the Jilin Province rice fields (Jilin, 2021) and the full straw return practices of local farmers, the amount of straw incorporated through both treatments (T1 and T2) was approximately 8,000 kg ha−1. Air-dried straw was chopped into 7.0–10.0 cm segments, evenly spread over the plots, and then flooded to a water depth of 3.0–5.0 cm. A rotary tiller (Model 1JH-150, Mengcheng County Yutian Machinery Co., Ltd., Bozhou, China) was used to incorporate the straw into the 5.0–20.0 cm soil layer.

Following the guidance of previous research (He et al., 2024), a compound fertilizer (N:P2O5:K2O = 17:17:17) was applied at 650.0 kg ha−1 as a basal dressing before transplanting. Topdressing with urea (N ≥ 47%) was applied at 30.0 kg ha−1 during the tillering stage and at 20 kg ha−1 during the heading stage. The total nutrient inputs were 150.0 kg ha−1 of N, 110.0 kg ha−1 of P2O5, and 110.0 kg ha−1 of K2O.

Rice seedlings were transplanted on May 27, 2022, and May 29, 2023, at a density of 16.7 hills m−2, with 3.0–4.0 seedlings per hill. A water depth of 3.0–5.0 cm was maintained from transplanting until approximately 30.0 days after heading. The fields were drained on September 13, 2022, and September 16, 2023. Pests, weeds, and diseases were strictly controlled throughout the experiment to prevent biomass and yield losses.

Sampling and measurements

Meteorological data

Temperature and sunshine duration data during the experimental period were provided by the Jilin City Meteorological Bureau.

Straw weight loss rate and breaking force

The weight loss rate and breaking force of the incorporated straw were measured using the nylon mesh bag method (Yan et al., 2019) from April 27 to September 16, 2022. We placed 1 kilogram of pre-dried rice straw (7.0–10.0 cm in length) into nylon mesh bags. For T1, the bags were buried at a depth of 15.0–20.0 cm on September 28, 2021, and for T2, they were buried on May 27, 2022. Bags were retrieved monthly from April to September 2022 during the latter part of each month. The straw was rinsed to remove soil particles, oven-dried at 75.0 °C and weighed. The rate of straw weight loss was calculated using the following formula: Strawweightlossrate%=SWi−SWfSWi×100

where SWi is the initial straw weight, and SWf is the weight at different sampling times.

Straw breaking force was measured using a plant stalk strength tester (Model YYD-1A, Zhejiang Top Cloud-agri Technology Co., Ltd., Hangzhou, China). Three replicates were conducted per treatment plot.

Rice bleeding sap rate measurement

The bleeding sap rate of rice was measured using a previously described method (Ansari et al., 2004), during the early, middle, and late parts of each month from July 3 to September 25, 2022. Representative plants from each treatment plot were cut 20.0 cm above the soil surface. Pre-weighed absorbent cotton was placed over the cut stem, covered with polyethylene film, and secured with a rubber band. The cotton was collected and weighed after 1 h, and the number of stems on each cut plant was recorded. The bleeding sap rate per stem was calculated using the following formula: Bleedingsaprateperstem=CWf−CWiSN

where CWi is the initial cotton weight, CWf is the cotton weight after 1 h, and SN is the number of stems. Three replicates were conducted per treatment plot.

Methane sampling and measurement

Methane emission fluxes (CH4) were measured using the static closed-chamber method (Ding, Jiang & Cao, 2021) during the early, middle, and late parts of each month from May 27 (transplanting) to September 16, 2022. Before transplanting, stainless-steel bases were installed in the sampling area, and two rice hills were planted inside each base. A transparent polypropylene chamber (40.0 cm × 30.0 cm × 100.0 cm) equipped with a thermometer and a small fan was placed over the base. Gas samples (100.0 ml) were collected at 0, 15, and 30 min between 9:00 and 11:00 AM on sampling days, ensuring no atmospheric exchange. The samples were stored in light-proof gas sampling bags and transported to a laboratory for analysis. Methane concentrations were determined using a gas chromatograph (GC-2014, Shimadzu, Kyoto, Japan), which was calibrated using certified methane standards (CH4 in N2, purity: 99.9%) at concentrations of 10.0 ppm and 100.0 ppm. Methane emission fluxes were calculated according to a previously described method (Guan et al., 2024). Three replicates were conducted per treatment plot.

Soil redox potential measurement

Soil redox potential (ORP) was measured around the static chambers using a soil ORP meter (Model TR-901, INESA, Shanghai, China) at the same time the methane was sampled. Three replicates were conducted per treatment plot.

Agronomic trait measurement

The intensive monitoring of dynamic agronomic traits was conducted primarily in the 2022 growing season. Plant height, stem number per hill, and chlorophyll content (measured using an SPAD-502 PLUS meter, Konica Minolta, Tokyo, Japan) were recorded at 15.0 consecutive hills in each treatment plot approximately every 15 days from June 5 to September 20, 2022. Three replicates were conducted per treatment plot.

Soil sampling and analysis

Soil samples (0.0–20.0 cm depth) were collected from the experimental paddies after the rice harvests in autumn 2022 and 2023 using a five-point sampling method, with sampling points spaced 5.0–10.0 m apart. Samples from each plot were thoroughly mixed to form a composite sample of approximately 500.0 g. Visible impurities were removed, and the soil was air-dried and passed through an 8.0 mm mesh sieve before analysis. The soil’s nutrients and chemical properties were determined as follows: alkali-hydrolyzable nitrogen (N) was analyzed using the Kjeldahl method (Bao, 2000), and soil organic matter was examined using the K2Cr2O7-H2SO4 oxidation method (Bao, 2000). Additionally, available phosphorus (P) was examined through extraction with 0.5 M NaHCO3 and measurement via molybdenum-antimony colorimetry (Bao, 2000); available potassium (K) through extraction with 1.0 M ammonium acetate and measurement via flame photometry (Bao, 2000); and exchangeable calcium (Ca) and magnesium (Mg) through extraction with ammonium acetate and measurement via atomic absorption spectrophotometry (Bao, 2000). Three replicates were conducted per treatment plot.

Yield determination

Yield components were measured for each treatment plot on September 29, 2022, and September 28, 2023. Yields were calculated based on the seed-setting rate, thousand-grain weight, number of grains per panicle, and number of panicles per square meter. Three replicates were conducted per treatment plot.

Statistical analysis

The experimental data were collated and initially processed using WPS software (Version 365, Kingsoft Office Software, Co., Ltd., Zhuhai, China). We performed analysis of variance (ANOVA), correlation analysis, and significance tests for differences between years, treatments and treatment × year interactions using the standard least squares method in JMP software (Version 18.0 Pro, SAS Institute, Cary, NC). Differences between treatment means were assessed using Tukey’s honestly significant difference (HSD) test with a significance level of p < 0.05. Figures were generated using Origin software (Version 2025, OriginLab Corporation, Northampton, MA).

Results

Climate conditions

Figure 1 shows the changes in air temperature and sunshine duration at 10-day intervals during the experimental period. The mean temperatures during most intervals in 2022 and 2023 were generally higher than the 2018–2023 average (referred to as “normal”). In 2022, the temperature fluctuations were more pronounced, with peak temperatures occurring in late July and early August. In 2023, the temperatures during most intervals remained consistently above normal. Temperatures after the heading stage in mid-August were significantly higher than normal. Specifically, early April and late October were 3.3 °C and 2.4 °C warmer than the corresponding normal periods. The only significantly lower-than-average temperature occurred in late April, which was 3.4 °C below normal.

Figure 1 Mean air temperature and sunshine duration at 10-day intervals during the experimental period.

(Normal) The 5-year average form 2018 to 2023. (E) Early part of the month. (M) Mid part of the month. (L) Late part of the month.

The total sunshine duration in 2022 was higher than normal but exhibited greater overall fluctuations, especially from late August onward. In contrast, the total sunshine duration in 2023 was lower than normal, displaying a clear downward trend from early July to early August. In particular, during early August, the sunshine duration was only 17.3 h, which was significantly lower than normal.

CH4 and ORP

Figure 2 shows the CH4 and ORP measurements taken during the experimental period in 2022. Significant negative correlations were observed between CH4 and ORP across all treatments, with correlation coefficients of −0.819 for CK (p < 0.01), −0.730 for T1 (p < 0.01), and −0.590 for T2 (p < 0.05). Significant differences in CH4 were observed between the treatments (p < 0.05). CH4 from CK remained relatively stable throughout the experimental period, reaching a peak of 10.8 mg m−2 h−1 on August 3. Before August 3, CH4 from CK was significantly lower than that observed during the other treatments (p < 0.05). T1 emissions peaked at 34.8 mg m−2 h−1 on June 28. Moreover, from June 6 to July 15, CH4 from T1 was significantly higher than those from both CK and T2 (p < 0.05). T2 emissions reached the first peak of 16.8 mg m−2 h−1 on June 28. After a slight decline, a second peak of 16.4 mg m−2 h−1 occurred on July 27. The second peak showed no significant difference from the T1 emission level (16.2 mg m−2 h−1) on the same date, but it was significantly higher than the value in CK (p < 0.05).

Figure 2 Dynamics of CH4 and ORP under different treatments in 2022.

Different lowercase letters indicate significant differences (p < 0.05) according to Tukey’s HSD test. (CH4) Methane emission fluxes. (ORP) Soil redox potential.

The ORP for all treatments exhibited a similar trend-initial decline followed by an increase during the experimental period-with significant differences observed between the treatments (p < 0.05). In CK, the ORP declined continuously from May 27 to June 28. After a slight increase, it dropped again, reaching a minimum value of −233.7 mV on August 14, which was 11 days after the CH4 peak in CK. From May 27 to July 15, the ORP values in CK were significantly higher than those in T1 and T2 (p < 0.05). The ORP dynamics for T1 and T2 were similar in the early phase of the experiment. Both treatments reached their minimum ORP values on July 3, 5 days after the first CH4 peak occurred on June 28, with −298.3 mV for T1 and −245.7 mV for T2. Starting on July 3, the ORP of T1 was significantly lower than that of T2 (p < 0.05). However, the ORP of T2 showed a brief decline on August 14, 18 days after its second CH4 peak. From then until September 16, no significant differences in ORP were observed between T1 and T2.

Straw decomposition

Figure 3 presents the changes in straw weight loss rate and breaking force after incorporation under different treatments in 2022. Significant negative correlations were observed between straw weight loss rate and breaking force for all treatments, with correlation coefficients of −0.983 for T1 (p < 0.001) and −0.941 for T2 (p < 0.01). Straw weight loss rates differed significantly between T1 and T2 (p < 0.05), with T2 exhibiting a consistently higher rate.

The weight loss rate of the T1 straw gradually increased from June 28 onward, showing substantial growth between June and July, reaching 38.7% by September 16. In contrast, the T2 straw began losing weight as early as April 27 (9.3%), with the loss rate progressively increasing to 41.6% by September 16.

Breaking force values also varied significantly between treatments (p < 0.05), with T1 demonstrating substantially higher values than T2. The breaking force of the T1 straw gradually decreased from May 27 onward, mirroring its weight-loss pattern with a pronounced decline from June to July, reaching 94.0 N by September 16. In comparison, the breaking force of T2 started decreasing on April 27 and maintained a relatively consistent downward trend throughout the growing season, declining to 55.0 N (58.5% of T1) by September 16.

Agronomic traits

Figure 4 illustrates the dynamics of bleeding sap rates under different treatments during the 2022 experimental period. Prior to July 27, no significant differences in bleeding sap rates were observed between treatments. After July 27, CK exhibited a progressively increasing trend, ultimately demonstrating significantly higher values than both T1 and T2 until September 2 (p < 0.05). The bleeding sap rate of CK peaked at 0.311 g stem−1 h−1 on August 27. At that time, the values for T1 and T2 were 0.158 g stem−1 h−1 and 0.171 g stem−1 h−1, respectively, representing only 50.8% and 55.0% of the CK value.

Figure 3 Straw weight loss rate and breaking force under different treatments in 2022.

Different lowercase letters indicate significant differences (p < 0.05) according to Tukey’s HSD test.

Figure 4 Dynamics of bleeding sap rates under different treatments in 2022.

Different lowercase letters indicate significant differences (p < 0.05) according to Tukey’s HSD test.

T2 exhibited greater fluctuation than T1 in bleeding sap rates. Although no significant differences were observed between T1 and T2 before August 27, T2 consistently maintained slightly higher values than T1. All treatments showed pronounced declines in bleeding sap rates after the field was drained on September 13. Notably, T1 displayed a transient increase post-drainage, a trend not observed in T2.

Figure 5 shows the changes in plant height, stem number, and leaf color across different treatments during the 2022 experiment. No significant differences in plant height were observed between the treatments. The plant height increased steadily throughout the growth period, with the rate of increase accelerating after June 22. This growth gradually slowed down after August 4, entering a plateau phase after September 4. On September 20, the final plant heights were 129.1 cm for CK, 129.4 cm for T1, and 125.3 cm for T2.

Figure 5 Growth period characteristics in 2022.

Different lowercase letters indicate significant differences (p < 0.05) according to Tukey’s HSD test.

Stem number increased in all treatments beginning on June 5, with a rapid increase after June 22 and a peak on July 8. At this time, there were 546.1 stems m−2 with CK, 472.6 stems m−2 with T1, and 518.8 stems m−2 with T2. There were significantly more stems in CK than in T1 (p < 0.05). Stem numbers gradually declined. However, except on July 22, the stem number in CK remained significantly higher than in T1 (p < 0.05). On September 20, the final stem numbers were 371.3 stems m−2 for CK, 345.2 stems m−2 for T1, and 361.3 stems m−2 for T2.

The trend in leaf color changes was similar to that of stem numbers across treatments. Leaf color increased rapidly starting on June 22, peaked on July 8, and then gradually declined. On August 4, the Soil Plant Analysis Development (SPAD) values were 31.5 for CK, 29.0 for T1, and 31.9 for T2, with T2 significantly higher than T1 (p < 0.05). On September 20, the final SPAD values were 22.2 for CK, 20.7 for T1, and 21.6 for T2, with CK significantly higher than T1 (p < 0.05).

Soil properties

Figure 6 shows the differences in soil properties between the treatments during the 2022 and 2023 trials. No significant year-to-year variation was observed in any soil properties. Additionally, K and organic matter content showed no significant differences between treatments. N under T2 was 175.7 mg kg−1 in 2022 and 176.7 mg kg−1 in 2023, while N levels under CK were 168.0 mg kg−1 in 2022 and 164.9 mg kg−1 in 2023. N was significantly higher under T2 than under CK (p < 0.05), with increases of 4.6% and 7.2%, respectively. P under T2 was 26.9 mg kg−1 in 2022, significantly higher than the value of 23.3 mg kg−1recorded under CK (p < 0.05). Ca also showed significant differences between treatments in both 2022 and 2023 (p < 0.05). In 2022, the Ca level under T2 was 26.4 cmol kg−1, which was significantly higher than the 25.5 cmol kg−1observed under CK (p < 0.05). In 2023, Ca ranked from highest to lowest as follows: 26.7 cmol kg−1under T2, 26.2 cmol kg−1under T1, and 25.1 cmol kg−1under CK. In 2022, the Mg content was 9.5 cmol kg−1 under T2 and 9.3 cmol kg−1 under T1, with T2 significantly higher than T1 (p < 0.05).

Figure 6 Soil properties under different treatments in 2022 and 2023.

Different lowercase letters indicate significant differences (p < 0.05) according to Tukey’s HSD test. N, Nitrogen; P, Phosphorus; K, Potassium; OM, Organic matter; Ca, Exchangeable calcium; Mg, magnesium; T, Treatment; Y, Year; T×Y, The interaction effect between treatment and year. (**, ***) Significant differences at the 1% and 0.1% levels, respectively. NS, No significant difference.

Yield and yield components

Figure 7 shows the yields and yield components under different treatments during 2022 and 2023. Panicle numbers showed no significant differences between treatments or years. However, the mean panicle numbers for 2022 and 2023 were 355.9 m−2 for CK, 328.1 m−2 for T1, and 340.4 m−2 for T2. The number of panicles under CK was significantly higher than that under T1 (p < 0.05).

Figure 7 Yield components and yield under different treatments in 2022 and 2023.

Different lowercase letters indicate significant differences (p < 0.05) according to Tukey’s HSD test. (PN) P anicle number. (NGP) N umber of grains per panicle. (SSR) Seed setting rate. TGW, 1,000-grain weight; T, Treatment; Y, Year; T×Y, The interaction effect between treatment and year. (*, ***) Significant differences at the 5% and 0.1% levels, respectively. NS, No significant difference.

In 2022, the number of grains per panicle was 116.8 for CK, 108.6 for T1, and 114.3 for T2. Both CK and T2 had significantly more grains per panicle than T1 (p < 0.05). In 2023, the corresponding values were 113.5 for CK, 108.6 for T1, and 110.4 for T2. CK had significantly more grains per panicle than both T1 and T2 (p < 0.05). The mean number of grains per panicle over the two years was 115.2 for CK, 108.6 for T1, and 112.3 for T2. Both CK and T2 had significantly higher mean values than T1 (p < 0.05).

No significant differences were observed between treatments in seed setting rate or thousand-grain weight. However, the mean seed setting rate across treatments was 93.3% in 2022 and 92.4% in 2023. The seed setting rate in 2022 was significantly higher than that in 2023 (p < 0.05).

The yields under CK were 9,412.9 kg ha−1 in 2022 and 8,990.5 kg ha−1 in 2023, both of which were significantly higher than under T1 (8,171.5 kg ha−1 in 2022 and 7,814.0 kg ha−1 in 2023, p < 0.05). The mean yield over the two years was 9,201.7 kg ha−1 for CK, 7,992.8 kg ha−1 for T1, and 8,622.9 kg ha−1 for T2. Significant differences in yield were observed between treatments (p < 0.05).

Discussion

Methane emissions and ORP

Compared to the approach in which no straw is returned, the strategy of incorporating rice straw can significantly increase methane emissions (Hou et al., 2013). Based on previous research, which indicates that methanogen activity in paddy systems typically ceases below approximately 10.0 °C (Conrad, 2023), the ambient environmental temperatures at this study site remained consistently below 10.0 °C from mid-October 2021 to late April 2022 (Fig. S1). Crucially, the soil remained frozen from November 2021 through March 2022. Consequently, methane emissions during this extended period are inferred to have been negligible. In this study, methane emissions differed significantly between the treatments (Fig. 2). Previous research has shown that because methanogen activity is temperature dependent, methane emissions from ecosystems such as paddy fields significantly increase with seasonal rises in temperature (Yvon-Durocher et al., 2014). Furthermore, returning straw leads to an earlier peak in methane emissions compared to treatments with no straw incorporation (Hu et al., 2016). The results of this study are consistent with these trends. T1 and T2, which implemented straw return, exhibited substantial methane emissions during the warmer months of June and July, with peak emissions occurring on June 28. In contrast, the peak methane emission for CK, which involved no straw return, occurred in early August when temperatures were highest (Fig. 1A). This resulted in concentrated methane emissions during the early warm season. The peak methane emission value for T1 was 2.1 times that of T2 and 3.2 times that of CK (Fig. 2). Water management during fallow periods has also been shown to significantly increase methane emissions (Sander et al., 2018). In this study, the total methane emissions over the entire growth period were 62.1 mg m−2 for CK, 183.0 mg m−2 for T1, and 123.9 mg m−2 for T2. The total emissions from CK were approximately 33.9% of those from T1, while the emissions under T2 were approximately 67.7% of T1. This difference occurred because T2 involved incorporating the straw using rotary tillage immediately after the autumn harvest in 2021, whereas under T1, it was incorporated in late May 2022. Previous studies in Inner Mongolia, China, demonstrated that flooding paddy fields after autumn straw incorporation maintained soil temperatures 1.0–2.0 °C higher than ambient environmental temperatures until the following May (Wang et al., 2023a). Research indicates that the minimum temperature for straw decomposition under natural conditions is approximately 7.0 °C (Yan et al., 2019). Compared to T1, under which straw was incorporated in late May 2022, T2 involved extending the decomposition period after incorporation by about seven months. During this extended period, ambient environmental temperatures exceeded 7.0 °C for more than 50 days (Fig. S1), while soil temperatures were higher, greatly accelerating straw decomposition rates after incorporation. Previous research in the Tohoku region of Japan also confirmed that elevated soil temperatures during the season in which rice is not grown promoted rice straw decomposition and reduced methane emissions in the subsequent rice-growing season (Tang et al., 2016).

Concurrently, returned rice straw serves as a substrate for methane production (Ye et al., 2015), and its degree of decomposition is closely correlated with methane emissions (Contreras et al., 2012; Yuan, Pump & Conrad, 2014). In the present study, the straw lost weight earlier (April 27) under T2, and the weight loss rate throughout the growth period was significantly higher for T2 than for T1 (Fig. 3). A significant negative correlation was observed between the straw’s weight loss rate and breaking force. The breaking force under T2 was consistently lower than that under T1 throughout the growth period. These findings collectively demonstrate that T2 accelerated straw decomposition by extending the decomposition period after straw incorporation. Under T1, following incorporation on May 25, both the weight loss rate and breaking force of the straw changed rapidly during the warmer months of June and July as temperatures gradually rose. This also led to concentrated methane emissions from T1 during this period (Fig. 2). Between June 16 and July 27, methane emissions from T1 totaled 135.1 mg m−2, accounting for 73.8% of T1′s total methane emissions during the growth period. This amount was approximately 5.8 times greater than that of CK and 1.9 times greater than that of T2 during the same period.

Consistent with previous research (Huang et al., 2005), a significant negative correlation between methane emissions and ORP was observed in this study. Previous studies have demonstrated that carbon input reduces ORP (Mattila, 2024) and that methane emissions are closely linked to ORP, with straw incorporation accelerating the decline in redox potential (Minamikawa & Sakai, 2006). Therefore, the concentrated methane emissions from T1 likely caused a substantial decrease in ORP during this period. The lowest ORP value observed for T1 was significantly lower than that of T2. The mean ORP values during the growth period were −148.2 mV for CK, −175.9 mV for T1, and −155.5 mV for T2. The soil under T1 exhibited stronger reducing conditions than it did under the other treatments.

Soil nutrients

Although numerous studies have reported that straw return increases the total organic carbon content in soil (Jin et al., 2020; Nakajima et al., 2016; Wang et al., 2017), the results from our two-year experiment did not corroborate this trend (Fig. 6D). This may be attributed to the relatively short straw incorporation duration in this study, which likely contributed to the lack of significant differences between years and treatments. Previous research has shown that soil’s total nitrogen content rises significantly following autumn puddling and rotary tillage incorporation compared to treatments with no straw return (Wang et al., 2023b). In this study, N under T2 was significantly higher than that under CK in both 2022 and 2023 (Fig. 6A).

Furthermore, other studies have shown that straw return can increase soil’s phosphorus content by 39.7% over the levels observed with no straw return (Jin et al., 2020). In this study, the P and Ca levels under T2 in 2022 were significantly higher than those under CK, while T1 did not exhibit such an increase (Figs. 6B, 6E). As mentioned previously, this can be attributed to the fact that in T2, irrigation was applied immediately after straw incorporation in the previous year. This allowed decomposition to begin earlier than in T1. By late April 2022, before transplanting, nearly one-tenth of the straw in T2 had already decomposed. Furthermore, on September 16, the weight loss rate in T2 was 2.9% higher than in T1 (Fig. 3). This enabled more nutrients from the decomposed T2 straw to be released into the soil, thereby increasing the soil’s nutrient content.

Agronomic traits

The dynamic agronomic traits were intensively monitored during 2022 to capture temporal responses of rice growth to straw incorporation. Although these traits were not measured over both years, the key yield and soil nutrient outcomes remained consistent across 2022 and 2023 (Figs. 6 and 7), supporting that the agronomic patterns observed in 2022 were representative of treatment effects under continuous practice. A decrease in ORP increases anaerobic respiration in rice roots (Tolley, De Laune & Patrick, 1986). Waterlogged soil generates toxins at lower redox potentials, inhibiting plant growth (Tokarz & Urban, 2015). It does this primarily through soil phytotoxins and other reductive byproducts that cause severe stress to plant roots (Pezeshki & De Laune, 2012). In this study, differences in ORP between treatments also affected sap exudation rates—an indicator of root activity—with significant differences observed across treatments (Fig. 4). The sap exudation rate in CK was significantly higher than those in T1 and T2. Research shows that the morphological and physiological traits of rice roots are negatively correlated with methane emission flux (Chen et al., 2019). In this study, during June and July—the period in which the most concentrated methane emissions and the lowest ORP values were observed (Fig. 2)—the sap exudation rates across all treatments, including CK, remained relatively low. Although the sap exudation rates in T1 and T2 increased slightly 13–14 days after reaching their lowest ORP values, they remained significantly lower than those in CK. This suggests that the lower ORP associated with straw return imposed persistent stress on the rice roots.

Previous studies suggest that straw return treatments can increase the height of rice plants and the number of stems they produce (Luo et al., 2024). However, our study found no significant differences in plant height between the treatments (Fig. 5A). Additionally, following the peak in stem numbers on July 8, CK consistently exhibited significantly higher stem numbers than T1, except on July 22 (Fig. 5B). As discussed above, stem number was affected by treatment-specific variations in ORP. T1 experienced stronger soil reducing conditions from July 3 to August 3 than CK and T2 (Fig. 2), leading to a reduced formation of lower-position tillers (Gao, Tanji & Scardaci, 2004), thereby decreasing the total stem number. In contrast, T2 maintained higher ORP than T1, particularly after July 27. Moreover, as previously noted, the enhanced straw decomposition observed under T2 (Fig. 3) facilitated a greater release of nitrogen into the soil, improving the availability of nutrients (Fig. 6A). As a result, although T2 experienced reductive stress (resulting in fewer stems than CK), the difference was not statistically significant. Owing to T2′s higher sap exudation rate (Fig. 4) and improved soil nutrient conditions (Fig. 6) by August 3, its leaf color values were significantly higher than T1 at the heading stage on August 4 (Fig. 5C).

Yield and yield components

An increase in ORP promotes a higher panicle number in rice, thereby enhancing yields (Minamikawa & Sakai, 2006). In this study, CK exhibited a significantly higher panicle number than T1 (Fig. 7A), which is consistent with CK’s trend of higher stem numbers throughout the growth period (Fig. 5B). Research conducted in Jiangsu, China, has shown that returning rice straw increases total nitrogen uptake, particularly during the panicle differentiation stage, thereby promoting the development of larger panicles and potentially higher yields (Li et al., 2023b). However, as previously stated, the significant decrease in ORP under T1 induced pronounced reductive stress. This resulted in significantly higher leaf color values for CK and T2 than T1 during the late growth stage; a higher sap exudation rate was also observed for CK than T1 after heading in early August (Fig. 4). As a result, both CK and T2 had significantly more grains per panicle than T1 (Fig. 7B). The mean seed-setting rate across treatments in 2023 was significantly lower than in 2022. This decline may be attributed to the high temperatures that began in mid-August 2023 (Fig. 1), which caused high-temperature-induced sterility (Fahad et al., 2019) and reduced the seed-setting rate (Fig. 7C).

Although numerous studies have demonstrated that the straw return method can enhance rice yields (Hou et al., 2021; Li et al., 2023b; Wang et al., 2015), treatment T1 in this study yielded a significantly lower yield than CK. While the yield under T2 was slightly lower than that of CK, no significant difference was observed (Fig. 7E). The underlying reasons can be explained as follows: different straw return treatments reduced ORP, impairing root activity and lower-position tillering capacity, thereby limiting panicle formation and reducing yields. However, compared to T1, T2 extended the straw decomposition period, improving the efficiency of the decomposition. This mitigated the adverse effects of soil reduction while enriching the nutrients in the soil. As a result, T2 maintained higher leaf color values than T1 during the late growth stage, supporting better grain formation per panicle and reducing the yield loss.

Conclusions

The results of this study demonstrate that T2 offers significant advantages over T1 for managing rice straw in cold-region. A significant negative correlation between CH4 and ORP was observed following the straw return process, with the impact of CH4 on ORP exhibiting a temporal lag. Crucially, T1 led to rapid straw decomposition, which coincided with high summer temperatures. This triggered concentrated CH4 total emissions 47.7% higher than those of T2, substantially decreasing ORP and inducing severe soil reduction stress. This stress impaired rice root activity, which was indicated by lower bleeding sap rates and stem numbers, and it ultimately resulted in a 12.7% lower yield than that of CK.

In contrast, T2 extended the straw decomposition period, resulting in a 2.9% higher final weight loss than T1. This effectively distributed the decomposition over a longer timeframe and prevented the intense, concentrated CH4 emissions observed under T1 during the summer. Consequently, T2 caused a smaller decline in ORP, significantly alleviating soil reduction stress compared to T1. Furthermore, the enhanced decomposition effectuated under T2 facilitated a greater release of nutrients from the straw, significantly increasing the levels of N, P, and Ca in the soil compared to CK, thereby improving the nutrient supply and boosting crop growth.

While the grain yield under T2 exhibited no significant differences from that under CK, the yield under T1 was significantly reduced. This demonstrates that incorporating rice straw via puddling and rotary tillage immediately after the autumn harvest is a superior strategy to adopt in China’s cold-region rice systems. This practice effectively extends decomposition times, alleviates soil reduction stress, promotes more complete decomposition, enhances key soil nutrient levels, and reduces peak CH4 emissions during the critical summer period—developments associated with led to yield stability. Based on these benefits, the incorporation of puddling in autumn demonstrates strong potential for productive integration into local carbon-neutral farming policies.

Supplemental Information

Supplemental Information 1 Raw data

Supplemental Information 2 Temperature from post-autumn straw incorporation to March of the following year

(E) Early part of the month. (M) Mid part of the month. (L) Late part of the month.

We thank Yongji County Jiuyuefeng Family Farm (Jilin, China) for providing all the rice materials used in this study.

Additional Information and Declarations

Competing Interests

Author Contributions

Data Availability

The authors declare there are no competing interests.

Bing He conceived and designed the experiments, performed the experiments, authored or reviewed drafts of the article, and approved the final draft.

Yonglin Si performed the experiments, analyzed the data, prepared figures and/or tables, and approved the final draft.

Chao Li performed the experiments, analyzed the data, prepared figures and/or tables, and approved the final draft.

Limin Wang performed the experiments, authored or reviewed drafts of the article, and approved the final draft.

The following information was supplied regarding data availability:

The raw data is available in the Supplemental Files.

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
