# Peer review of "Accelerated straw decomposition and mitigated methane emissions via autumn puddling incorporation enhances soil health and yield stability in cold-region rice systems of China"

_PeerJ, doi:10.7717/peerj.20264_

## Round 0.1 · original submission · Minor Revisions

You are requested to do the revisions immaculately.

**Language Note:** The review process has identified that the English language must be improved. PeerJ can provide language editing services - please contact us at [email protected] for pricing (be sure to provide your manuscript number and title). Alternatively, you should make your own arrangements to improve the language quality and provide details in your response letter. – PeerJ Staff

·

Basic reporting

The manuscript is generally well-written and professionally formatted. The introduction is comprehensive and provides appropriate background on the issues of rice straw management and methane emissions in cold-region rice systems. The figures are relevant and of good quality, with sufficient resolution and informative captions. The authors cite recent and relevant literature to support the context and need for the study.
Abstract
• Problem statement is missing in the abstract
• Line 28- 29 sentences is confusing. It can be write as T2 shown significantly increased soil nitrogen, phosphorus, and calcium levels and maintained higher root activity and
• stem numbers compared to T1 and CK
• The authors must highlight some results of nitrogen etc
• The methane emission measured data is missing
• Line 31-34 abstract conclusion is not clear
Introduction
• No data is provided for decomposition of residues
• Consistency in hyphenation (e.g., “cold-region rice systems” vs “cold region rice systems”) should be improved.
• Objectives are not clear
• Need English language improvement

Experimental design

• Replication details could be clarified. While “three replicates” are mentioned, the spatial layout (e.g., randomization, blocking) and potential confounding environmental variables (e.g., soil microvariability) are not described.
• The methodology around straw decomposition temperature thresholds (e.g., 7°C minimum for decomposition) relies heavily on literature (Yan et al. 2019) and assumptions. It would be helpful to discuss any soil temperature monitoring that validates decomposition activity during the fallow period.

Validity of the findings

• The conclusion that T2 mitigates methane emissions needs to account for total CH₄ budgets more explicitly, especially including potential off-season emissions after autumn incorporation. This is critical if the field remained flooded after T2 incorporation, which could bias CH₄ emission savings.
• Soil organic matter (SOM) showed no significant increase across treatments, which conflicts with much of the cited literature. The authors acknowledge this but do not explore potential explanations (e.g., short study duration, high SOM baseline).
• Although differences in ORP are discussed, the causal link between ORP and yield components remains associative. More caution in drawing mechanistic conclusions is advised (e.g., “led to yield stability” vs “associated with reduced yield loss”).

Additional comments

Conclusion
• It is not written clearly
References
• Add DOI of the available references

Reviewer 2 ·

Basic reporting

1. Language and Structure
The manuscript is written in professional, clear English and complies with academic writing standards.
2. Figure Quality
All figures (Figures 1-7) are well-prepared with clear annotations. It is recommended that:
In Figure 2, the units for CH4 and ORP on the axes should be standardized to international units (e.g., mg CH4 m-2 h-1).
3. Data Completeness
The authors have provided the raw data files, which align with PeerJ policies.

Experimental design

4. Innovation and Necessity
This study presents the first systematic comparison of the ecological effects of straw incorporation in spring and autumn in cold-region rice fields in China, filling a research gap in straw management for high-latitude rice cultivation at 43°N. It is recommended to:
Add a comparative analysis with relevant agricultural systems abroad in the Introduction (lines 60-75).
Supplement the basic physicochemical properties of the soil and the carbon-to-nitrogen (C/N) ratio of the straw.
5. Methodological Rigor
The monthly sampling frequency effectively captures the dynamics of methane emissions.

Validity of the findings

6. Data Reliability
The temporal correlation between methane emission peaks and straw decomposition rates (lines 236-252) is biologically reasonable and consistent with the model predictions of Tang et al. (2016).
7. Statistical Methods
The Tukey’s HSD test (line 186) is appropriate for multiple group comparisons. However, it is recommended to:
Supplement significant differences with the corresponding ANOVA p-values (p < 0.05) where applicable.

Additional comments

The manuscript requires moderate revisions before acceptance, primarily focusing on refining statistical analyses and mechanistic discussions. This study holds significant academic value and practical implications for the sustainable management of temperate rice cropping systems.

·

Basic reporting

No comment

Experimental design

No comment.

Validity of the findings

No comment.

Additional comments

- The experimental design method has not been specified in detail, nor has it been clarified which specific layout type was employed.
- The basis for selecting a straw incorporation rate of approximately 8,000 kg ha⁻¹ is not justified in the manuscript. It is unclear whether this value was derived from actual field measurements, local farmer practices, or references.
- The methodology for determining agronomic parameters is lacking. Please explain why agronomic traits were measured only in one season!
- The methodology for CH4 measurement is lacking.
- Numerous inconsistencies remain in citations and units of measurement.

---

## Round 0.2 · Minor Revisions

The reviewer's comment about lack of detailed methodology for measuring agronomic traits and measuring them only in one season has been addressed in details in the rebuttal letter. However, a concise reasoning for this two-part comment should be included in the main manuscript as well. There is no need to specify the health reasons in the manuscript.

---

## Round 0.3 · accepted · Accept

The authors are commended for improving the article as per PeerJ standards.